# Distributed Learning Fractal Algorithm for Optimizing a Centralized Control Topology of Wireless Sensor Network Based on the Hilbert Curve L-System

**DOI:** 10.3390/s19061442

**Published:** 2019-03-23

**Authors:** Jaime Moreno, Oswaldo Morales, Ricardo Tejeida, Juan Posadas, Hugo Quintana, Grigori Sidorov

**Affiliations:** 1Escuela Superior de Ingeniería Mecánica y Eléctrica, Instituto Politécnico Nacional, Mexico City 07738, Mexico; oswmm2001@yahoo.com (O.M.); jpposadas@gmail.com (J.P.); jaimemor1979@hotmail.com (H.Q.); 2Escuela Superior de Turismo, Instituto Politécnico Nacional, Mexico City 07630, Mexico; ricardotp75@hotmail.com; 3Centro de Investigación en Computación, Instituto Politécnico Nacional, Mexico City 07738, Mexico; jaimemor1979@yahoo.com.mx

**Keywords:** Hilbert curve, WSNs, quality of service, centralized control topology, adaptive algorithm, L-system, swarm intelligence, optimization, distributed learning

## Abstract

Wireless sensor networks (WSNs) consist of a large number of small devices or nodes, called micro controller units (MCUs) and located in homes and/or offices, to be operated through the internet from anywhere, making these devices smarter and more efficient. Quality of service routing is one of the critical challenges in WSNs, especially in surveillance systems. To improve the efficiency of the network, in this article we proposes a distributed learning fractal algorithm (DFLA) to design the control topology of a wireless sensor network (WSN), whose nodes are the MCUs distributed in a physical space and which are connected to share parameters of the sensors such as concentrations of $CO_{2}$, humidity, temperature within the space or adjustment of the intensity of light inside and outside the home or office. For this, we start defining the production rules of the L-systems to generate the Hilbert fractal, since these rules facilitate the generation of this fractal, which is a fill-space curve. Then, we model the optimization of a centralized control topology of WSNs and proposed a DFLA to find the best two nodes where a device can find the highly reliable link between these nodes. Thus, we propose a software defined network (SDN) with strong mobility since it can be reconfigured depending on the amount of nodes, also we employ a target coverage because distributed learning fractal algorithm (DLFA) only consider reliable links among devices. Finally, through laboratory tests and computer simulations, we demonstrate the effectiveness of our approach by means of a fractal routing in WSNs, by using a large amount of WSNs devices (from 16 to 64 sensors) for real time monitoring of different parameters, in order to make efficient WSNs and its application in a forthcoming Smart City.

## 1. Introduction

The fourth industrial revolution (4IR) will encompass a wide range of technologies that are fusing the physical, digital and biological aspects, transforming the way in which human beings live, work and interrelate. In the 4IR, various technological fields will see major advances in the coming years that will affect all disciplines, economies and industries, as is the case of robotics, artificial intelligence, nanotechnology, quantum computing, biotechnology, the internet of things (IoT), 3D printing and autonomous vehicles. Artificial intelligence (AI) is the skill that allows machines to learn and adapt to different situations and problems. Due to its great versatility, the AI can work in conjunction with the IoT, allowing devices or micro controller units (MCUs) that are part of wireless sensor networks (WSNs) to be able to collect, process, and share data of different nature that flows through the network. It is expected that by 2025 26–30 million MCUs in homes and offices will be connected to the Internet, and that they will be equipped with sensors, processors and embedded software.

If we try to find an unique anthropomorphic sense, we have always thought that intelligence was something that belongs to a single entity, an individual with enormous complexity. Hence, AI projects have always been focused on generating supercomputers, capable of handling a lot of information at high speed. But, if that were not the idea and if we do not have to look for a single super-intelligent entity but something different, we can give a way to a collective intelligence as a distributed processing entity. For example, if we observe the large colonies of social animals such as bees, termites or ants, we see that, as individuals, each of its members is not so intelligent, just follows very simple algorithmic patterns. However, as a colony, as a group, their collective behavior is much more intelligent. An ant alone can not defend the colony or feed the queen, but many of them collaborate emerging as an intelligent entity.

Accordingly, we talk about the swarm intelligence defined by Beni andWang in [1], who tell us that it is the collective and decentralized behavior of self-organized systems in a natural or artificial way, being a basic principle of the intelligence of simple systems that can be transformed into complex systems. Swarm intelligence (SI) considers agents (biological or artificial) of a system or network as a collective or swarm that, without central control, collectively (unknowingly, and in a somewhat random way) perform tasks that normally require some form of intelligence.

The SI is related to the AI because it focuses on the study, modeling and prediction of collective or *swarm* behavior of individual technological units. The individual technological units, for example the MCUs, which make up WSNs, can be considered as a collective or swarm with intelligent interconnections to give rise to the emergence of different patterns at a global level, such as the optimization of bandwidth in the WSN network to make efficient the transmission of the parameters shared by all the nodes in the network.

The topology of a network is the structure or arrangement of the different elements or nodes that integrate it so that they are linked or connected. In WSNs the nodes can be the devices or MCUs and the links could be the bandwidth to transmit data throughout the network. The vast majority of physical WSNs display the star configuration, in which each one of the devices is connected to a central node or hub. In this topology if the hub fails, all the devices connected to it will be disconnected, so there is a high dependency of the system to the hub operation, that is, a failure in the hub would generate a global disconnection in WSNs. Part of this can be solved by using the AI and the SI, to be able to design topologies in which the nodes:learn and adapt to different situations and problem, andact as an intelligent swarm.

Thus, WSNs behave as a collective entity which does not depend on a central control or hub.

Therefore, it is necessary to employ a systemic paradigm that allows designing WSN topologies that are not subject to the performance of a single MCU or node, in order to ensure an optimal operation of these technological systems as a swarm, where the collective pattern that emerge are quality of service (QoS) requirements like end-to-end reliability and delay due to an efficient connectivity based on a bandwidth that links each of MCUs with their immediate neighbors at 300 Mbps, in such a way as to ensure that they are really going to share the parameters between all the nodes of WSNs when they are operating.

The synergy, or science of cooperation, is a systemic discipline that addresses the study of the behavior of systems made up of many individual parts that can generate complex collective behavior. These collective or swarm behaviors are accompanied by the emergence of functional structures, that is, special arrangements coordinated between the elements without an interference from an external agent and at different spatio-temporal scales, such as the phenomenon of self-organization. Thus, the main objective of synergy is to unify the principles for the study of systems in which self-organization can emerge. Therefore, the synergy allows us to use concepts, theories, methods, methodologies and models of various quantitative and qualitative disciplines to lead the study of systems, regardless of their nature such as swarm intelligence, artificial intelligence, fractal geometry and L-systems, among many others.

According to the above, in this paper was used the synergy paradigm to develop an effective algorithm for extending the range of transmission of a given WSN in an intelligent, adaptive and dynamic way under a swarm intelligence framework [2,3,4]. Specifically, we propose a reliable DLFA based on the Hilbert curve L-system, which considers the optimization of links in finding a path from a source to a destination by considering QoS constraints such as end-to-end reliability and delay. In this way, the main contributions of this work are as the follows:We model the optimization of a centralized control topology of WSNs.We propose a distributed learning fractal algorithm or DFLA to find the best two nodes where a device can find the highly reliable link between these node, in order to transfer or share parameters among a WSN.Through laboratory tests and computer simulations, we demonstrate the effectiveness of our approach by means of a fractal routing in WSNs. Thus, in our laboratory test we use a large amount of WSN devices (from 16 to 64 sensors) for real time monitoring of different parameters, in order to make efficient WSNs and its application in a forthcoming smart city.

The further sections of this article are organized as follows. First, we establish, in Section 2, we deeply report two state-of-the-art works in the reliable routing of WSNs and a brief summary of another well-known algorithms. In Section 3 and Section 4, an explanation of the theoretical definition of the integrated systems in the wireless sensor network and the Hilbert space curve, respectively, is given. The main scheme of this proposal is defined and analyzed in detail in Section 5. While in Section 6 the simulation and the comparison of the performances are announced, in order to verify the correction and feasibility of the proposal. Finally, the arguments and discussions of this work are analyzed in Section 7.

It is important to mention that this work is an extension of the article presented by Moreno et al. [5] in the Volume 11288 of the Lecture Notes in Artificial Intelligence, but we have completely modified the theoretical and methodological frameworks, in addition of the experimental results, since this proposal is a dynamic version of the static already presented algorithm.

## 2. Related Work

Energy efficiency has been one of the critical challenges in wireless sensor networks (WSNs) because the nodes in such networks have limited resources. Hence, several topology control protocols for WSNs have been proposed to decrease energy consumption of the nodes and increase the network capacity. For example, Javadi et al. in [6] proposed a topology control protocol based on learning automaton. The learning automaton of every sensor node chose the proper transmission range of the node using the reinforcement signal which was produced by the learning automaton of neighbor sensor nodes. They found that choice had a determinant impact on overall lifetime of the network. i.e., that feature could help the nodes to improve the energy efficiency of the network.

On the other hand, quality of service (QoS) routing is also another of the critical challenges in WSNs, especially for surveillance systems. Current multipath routing works can provision QoS requirements like end-to-end reliability and delay, but suffer from a significant energy cost. To improve the efficiency of the network with multiconstraints QoS parameters, Mostafaei in [7] modeled the reliable routing problem as an multiconstrained problem and proposed a distributed learning automaton (DLA) based algorithm to provision the QoS requirements for packet routing in WSNs. In terms of energy efficiency, this method tried to select best possible nodes to save other nodes’s residual energies by using a small number of sensors with high-reliable links to transfer information of a particular event in a network. According to him, his results demonstrated that his algorithm had a better performance than current state-of-the-art competitive algorithms in terms of end-to-end delay and energy-efficiency.

Both Javadi et al. and Mostafaei algorithms are specifically software-defined networking (SDN) in wireless sensor networks, i.e., SD-WSNs. Mostafaei and Menth in [8] define a SDN as a forwarding of elements are remotely confiured by centralized controllers instead through distributed control protocols. In this way, one of the goals of these SD-WSNs is being verified by every member of the WSN as a swarm or colony of small entities since there are controlled in a distributed way, but one of the drawbacks is that the configuration of the WSN is very complex.

In Figure 1, we depict the advances in software-defined wireless sensor networks. In this way, SD-WSNs can be sorted in nine classifications:Energy efficiency: sleep scheduling approaches are designed for switching the nodes into idle state if their functionality is not required. This classification can be reclassified as follows:Lifetime: possibility to utilize the nodes functionalities for a longer period of time.Coverage control: control activates or deactivates the sensor nodes to cover a network region.Clustering: nodes into clusters and there is a head node for each.Routing: transferring efficiently the network data.Mobility: physically moving the place of nodes due to external forces.Reliability: monitored data is transferred to the outside of the network via multi-hop connections.Quality of service: guaranteed level of service delivery to a network.Management: network configuration, provisioning, and maintenance.Localization: information of each node is necessary for many applications of WSNs.Wireless power transfer: a sensor node can transfer its energy to other nodes through an appropriate transmitter.Security: global overview of a device’s status in the network which results in identifying the malicious user and their activities.

For this reason and taking in to account these features, we propose a distributed learning fractal algorithm (DFLA) to design the topology of a wireless sensor network, whose nodes are the MCUs distributed in a physical space and which are connected to share different parameters of the sensors. We define the production rules of the L-systems to generate the Hilbert fractal, and we model the optimization of a centralized control topology of WSNs and proposed a DFLA to find the best two nodes where a device can find the highly reliable link between these nodes.

## 3. Wireless Sensor Networks: Devices and Interconnection Algorithms

In this section we expose the first part of the theoretical framework of this proposal since we define the most important features of the wireless sensor networks (WSNs) devices which would be the MCUs or nodes that shape our experimental wireless sensor network (WSN). On the one hand, the main characteristics of the ESP8266160 and Raspberry Pi 3 B+ embedded systems are exposed. On the other hand, these WSN nodes need to be reliable linked which is why WiFi Interconnection standard for WSNs is explained.

### 3.1. ESP8266

NodeMCU is a small WiFi plate ready to use in WSNs. It is mounted around the already known ESP8266 and exposes all its pins on the sides, see Figure 2. It also offers more advantages such as the incorporation of an integrated voltage regulator, as well as a USB programming port. It can be programmed through the Arduino IDE. It has an extensive community and documentation that will allow you to connect WSN to the outside world through a WiFi connection. Because it uses a USB CH340 converter, it is usually installed automatically by the operating system, although depending on the case, you may need to install the specific driver.

It is the availability of a WiFi connection in a microcontroller whose price is around $3 USD. Moreover, it is possible to program directly with the Arduino environment with what is the perfect chip to develop some WSNs. There are several models of ESP8266 that differ in the amount of available GPIOs and for technology of the WiFi antenna. In our case, we focus on the Model ESP-12-E because it is the most widely used, as well as the largest community.

The main features of the ESP8266 are the following:Processor: ESP8266 @ 80 MHz (3.3 V) (ESP-12E)4 MB of FLASH memory (32 MBit)WiFi 802.11 b/g/n3.3 V integrated regulator (500 mA)USB-serial CH340 converterAuto-reset function9 GPIO pins with I2C and SPI1 analog input (1.0 V max)4 mounting holes (3 mm)RESET buttonExternal power input VIN (20 V max)

As we can see, the simplicity of this WSNs device can be used for applications mainly of home automation or measurement of parameters that through the swarm intelligence can be used as a super organism which is simple in essence but powerful in practice.

### 3.2. Raspberry Pi 3 B+

From Figure 3, the Raspberry Pi is represented as one of the most important hardware solutions of the middle of the decade of 2010 and the field of WSNs. For a low price (US $35), users have the possibility to install their own mini-server, fully functional and enjoying an ideal power to meet their needs.

Since the first time it was launched, several updates of the Raspberry Pi have been presented, with an increase in its power, characteristics and for a very low price. Well, now it has joined the Pi series as the Raspberry Pi 3 Model B+. This model uses a Broadcom chip with an operating frequency superior to that of its predecessors, as well as incorporating the ability to operate with 64-bit operating systems.

Also, the Raspberry Pi 3 Model B+ incorporates a dual band WiFi card, which gives the feature of an improved transmission speed and better connectivity. Since 2012, when the Raspberry Pi was launched for the first time, this mini-computer has been overwhelmingly successful, surpassing even the expectations of its developers. In fact, in mid-2017 the company reported more than 15 million units sold around the world.

At first the Raspberry Pi was created as a mini-computer that would encourage children to increase their interest in the computer sector, especially for the ease of use. This same characteristic was the reason of its expansion towards other daily projects very different from the solutions proposed at the beginning by the Raspberry Pi Foundation.

All the features of WSNs embedded system Raspberry Pi 3 Model B+ are the following:CPU: Broadcom BCM2837B0,GPU: Cortex-A53 (ARMv8) 64-bit SoC @ 1.4 GHz,RAM: 1 GB LPDDR2 SDRAM,WiFi: 2.4 GHz and 5 GHz IEEE 802.11.b/g/n/ac,Bluetooth 4.2, BLE,Ethernet: Gigabit Ethernet over USB 2.0 (300 Mbps),40 pin GPIO,HDMI,4 USB 2.0 ports,CSI and DSI port to connect a camera and a touch screen,Stereo audio output and composite video,Micro SD, andPower-over-Ethernet (PoE).

The main difference of this new mini-computer with respect to previous models is that it includes an Integrated Heat Spreader (IHS) that increases 200 MHz frequency to SoC. As we have seen, the most notable improvement has been made in relation to wireless connectivity, where Cypress CYW43455 is given by Bluetooth 4.2 and WiFi.

The company has warned that the Raspberry Pi 3 Model B+ has a higher power consumption compared to previous models and it is recommended to use an adapter that provides up to 2.5 A.

The fields of application of the Raspberry Pi have been greatly expanded. Along with the numerous general usage possibilities for which the minicomputer is predestined, the realization of exceptional ideas is also proposed. To carry out projects with Raspberry Pi, certain prior knowledge is necessary, but if you have enough interest, there will be nothing to prevent the realization of your own projects. Quite the contrary, and is that the fact of experimenting with the board and learning new computer skills are the approach that lies behind the computer.

The internet is full of data on how to carry out numerous projects and applications for Raspberry Pi. The examples that follow provide a first impression of the possibilities offered by the minicomputer.

The versatility of this WSNs device, on the one hand, can generate generic applications such as web server, smart home unit, email server, VPN server, DNS server, 3D printers or recreational machines or video game consoles. But, on the other hand, applications for science such as the design and implementation supervisory control and data acquisition of the sedimentation process of water treatment plants [9], automated security modules for text and image steganography [10], surveillance robots [11], text readers for the visually impaired [12], or real time face detection [13].

As it is observed, the applications are diverse and with potential, that is why this WSNs device was chosen, given that it could enhance both connectivity or storage applications as well as research and scientific research.

### 3.3. WiFi Interconnection Standard for Wireless Sensor Networks

The WiFi Alliance has certified the standard that allows a connection without the need for an access point. The approval of WiFi Direct brings about a sensible improvement when communicating two devices with each other, to the point of reaching speeds much higher than Bluetooth 2.0, which implies great possibilities in the field of content synchronization.

WiFi Alliance, the organization that checks products with wireless network connections to ensure its proper functioning in the market, has announced that it certifies the first products with the WiFi Direct standard. It is a P2P connection between devices, bypassing the hitherto necessary access point. While waiting for hardware prepared for this type of communication to begin to flourish, there are already some brands such as Intel and Cisco that incorporate the standard in some of their products. From this moment it will be easier to share videos, images or music among several users, besides being much faster. The Bluetooth 2.0 connectivity, which could be presented as one of the best options for this task, is clearly exceeded in speed by the transfer rates offered by WiFi Direct. The synchronization of devices at the domestic level is another of the fields in which the new standard shows its usefulness, see Figure 4.

The growing expansion of mobile terminals leads to an increase in the use of applications and also in the transfer of data, which can be favored by this type of connection. The gamer sector, for example, will benefit, making it easier to connect several users to play online. WiFi Direct allows users to connect devices, when, where and how they want, and the WiFi Alliance’s certification program launches products that work well together, without taking into account the brand. There is no doubt about the progress that this implies, although it may be that the security debate—always attached to the news—will ignite. If the connection is left open it is possible for unknown users to access others, even if they notice. It must be remembered that not so long ago most of the domestic networks were unprotected.

## 4. Hilbert Curve L-System

Once the devices (MCUs) or nodes that make up the WNS were chosen, the next step was to define the mechanisms or rules through which these nodes would connect, based on the construction of the Hilbert fractal.

Benoit Mandelbrot coined the fractal term in 1975 to encompass all these phenomena that, due to fulfil all or some of the properties identified, had been excluded from a formal classification. Its definition was reduced to the comparison between two measurements of the dimension. The classic one, named of topological, and a new measure of the fractional dimension, which he called fractal. Fractal objects have a fractal dimension strictly greater than the topological dimension.

There is in nature a type of fractal, in two and three dimensions, where are curves that fill the plane or volume that contains them. Mandelbrot showed many examples of fractal structures found in nature [15], such as the vascular networks of trees, ferns, leaves and river deltas. Examples in animal biology include cancer networks in the lungs and blood vessel networks of the circulatory systems of animals [16].

The curves that fill the plane that contains them do so in a specific order by continually changing direction or passing through each point that is in the defined space [17]. There is a type of fill-space curves that satisfies the following conditions: (i) curves in a single layer that do not intersect, (ii) each point in the curve is at a constant distance unique to any other point, and (iii) there is only one starting point and one stopping point of the curve in a single layer. These curves have been widely studied more than a century ago in various fields such as mathematics, information processing, data consulting or image compression. The Hilbert fractal is a fill-space curve that satisfies these three criteria and it can be generated by using specific rewriting rules of the L-systems [18].

The L-systems, developed by the Hungarian botanist Aristid Lindmayer in 1968, emerged as an attempt to discreetly describe the development and growth of multicellular organisms. The objective was a formalization in which purely morphological observations were considered together with the genetic, cytological and physiological aspects. It was intended to propose abstract systems that would serve as a theoretical framework to study all these aspects [19]. The model (based on discrete and combinatorial mathematics) was opposed to traditional formalization based on continuous functions and differential equations. The process consists in discretizing both space and time and encoding the rules of behavior in a symbolic way.

The L-systems represent an evolution of other discrete mathematical models (such as von Newman’s auto-reproductive cellular automatons, or the neuronal means of McCulloch and Pitts) but, although they have in common enough of their mathematical objects, they differ in such important aspects as the type of growth allowed to the structure that the organism represents.

The L-systems consist of a set of axioms and rules used to generate recursive systems, using a parallel rewrite mechanism [18]. In the L-systems a word (axiom) is associated to the initiator, and production rules to the iterator, besides using an adequate graphic interpretation that allows us to obtain images when interpreting the words derived by a given alphabet.

For the graphic validation of the production rules in the L-Systems, ref. [20] developed a graphical interpretation algorithm to construct fractals, such as the Hilbert space-curve, in a very concise and compact way, that is, as L-systems with very few production rules. The Paper’s algorithm start with an object (in our case, the parameters to be shared among sensors) that, when moving in the system, generates a trace or curve. These parameters to be shared are fully trained to understand commands, in the form of symbols (from the list of symbols used in the L-system, such as the letters of the alphabet or the + or − signs):F, move forward by a certain fixed step length *l*,+, turn left (counterclockwise) by a fixed angle δ, and−, turn right (clockwise) by the angle δ.

The parameters to be shared must be able to interpret a single symbol according to a certain sequence of commands defined as *F*, + and −. Therefore, from this moment it is proposed to use the symbols L and R. The symbol L represents a kind of small deviation from a forward step forward (starts with a left turn) and the symbol R represents a small detour in the other direction (starts with a right turn).

The construction of a Hilbert fractal curve is presented in Figure 5, in which the dotted square shows the area to be filled by the curve. This square is divided into four squares. The construction begins with a curve $H_{0}$ that connects the centers of the quadrants through three line segments. It is assumed the size of the segment to be 1. In the next step four copies are generated (reduced by 12) of this initial stage and the four copies are placed in the four quadrants.

In this way the first copy is rotated clockwise and the last copy is rotated counterclockwise at π2 radians. The start and end points of these four curves are connected using three line segments (size 12) and the resulting curve is called $H_{1}$. In the second step, $H_{1}$ is scaled by 12 and the four copies are placed in the quadrants of the square as in step one. Again the start and end points are connected using three line segments (now of size Â14) and $H_{2}$ is obtained. This curve contains 16 copies of $H_{0}$, each of size 14. As a general rule, in step *n*, $H_{n}$ is obtained from four copies of $H_{n-1}$, which are connected by three line segments of length 12n, and this curve contains 4n copies (reduced by 12n ).

## 5. Distributed Learning Fractal Algorithm in a Wireless Sensor Network

The proposed methodology in this work is defined by the following stages:Hub MCU in the WSN,Identification of $LRN_{i}\left(t\right)$,Sorting by means of the Hilbert fractal production rules,Seed topology of the growing WSN.

It is also important considering the general parts of the environment of the proposal, so any configuration of WSNs consists of a wireless access point (WAP) and a β number of WSN nodes which can be increased or decreased in its amount. In this case β embedded systems or MCUs that composed the WSN, which shares parameters among sensors inside a given room or floor plan (see Figure 6), β ranges dynamically as follows:(1)$$1\geq \beta >4^{n}.$$

In this way, we also define $WSN_{N}$ by the Equation (Equation 2) as
(2)$$WSN_{N}\left(t\right)=\sum_{i\geq 1}^{\beta }MCU_{i}\left(t\right),$$
where $t=\left\{t_{0},t_{1},t_{2},\ldots t_{j}\right\}$, i.e., a specific period of time $\left(t\right)$ when the WSN is defined as a set of $MCU_{i}$ with at least one member up to $4^{n}$ members. Whereas *n* is the Hilbert Fractal level needed to go from the first to the last $MCU_{i}$ and it is expressed by Equation (Equation 3). Here *i* represents the index inside the WSN $\left(t\right)$ of a given MCU, namely $MCU_{1}$ is the first MCU, $MCU_{2}$ the second one and so on:(3)$$n=\left\lceil \log_{4}\left(\beta \right)\right\rceil .$$

It is important to emphasize that the WSN has β MCUs in the time frame $\left(t_{j}\right)$, but it can have up to $4^{n}-1$ devices. If $\beta \geq 4^{n}$, the value of *n* is recalculated by the Equation (Equation 3) and a new topology is reconfigured, otherwise the algorithm just adds a new node in the network, with the purpose that each node of the same network is linked or indexed by the curves that make up the Hilbert fractal. Subsequently, the effectiveness of the methodology exchange parameter throughout the network is measured. Finally, the point-to-point algorithms of WiFi (P2P) and SoftAP are configured.

### 5.1. First Stage: Hub MCU

By one hand, it is important to establish that $\beta (t_{j})$ WSN devices are randomly distributed along a certain floor plan which forms a matrix γ with $2^{n}\times 2^{n}$ dimension. Namely, our proposal estimates a certain topology according the β devices in the room in a certain period of time $t_{j}$. Then, the main WAP or hub is placed in the center of the main room, although it can be placed anywhere within the room, to have connections at any direction. That is, every connection to the WAP will be as the mainstream star topology.

In this way, all $\beta (t_{j})$ devices try to connect by the IEEE 802.11 b/g/n protocol to the nearest WAP, some will find others not. Those who find the only WAP establish a connection with it, while those who do not have a link to a central node or router enable the P2P WiFi mode in listening or waiting mode. All the nodes connected to the WAP are completely disabled from the WiFi Direct mode. Furthermore, WAP sends a broadcasting to the whole network to measure what is the link speed of each of the nodes that were connected to it. On the other hand, it is defined as MNI to the label of each device or micro controller network identifier, which has eight bits with which our system is limited to connect a maximum of $2^{8}$ development boards, that is, the fractal topology will only grow up to a fourth level.

In this first generation of MNI’s, a consecutive number will be assigned according to the speed of its link with the WAP. Thus, certain $MCU_{i}$ has the best link so this device is labeled as i=1 or $MCU_{1}$. Any floor plan had some walls, which attenuate the signal, that is why certain devices with shorter linear distance have a slower maximum speed than others that are further away from the WAP.

### 5.2. Second Stage: Identification of $LRN_{i}\left(t\right)$ and $LSN_{i}\left(t\right)$

Ones $MCU_{i=1}\left(t\right)$ is identified as the main node in the WSN. Hence, the WSNe(t) embedded systems, $MCU_{1}\left(t\right)$ included, enable WiFi Direct mode and full a table with the bandwidth of the nodes are next to them, in order to generate a list of reliable Nodes in a certain period of time *t* ($LRN_{i}\left(t\right)$). Every MCU generates a particular $LRN_{i}\left(t\right)$, then the proposed WSN can generate $4^{n}$ LNR’s at the instant *t*. In addition, every $LRN_{i}\left(t\right)$ contains the bandwidth of all $MCU_{i}\left(t\right)$ with which it establishes connection. In order to belong to the wireless sensor network every $MCU_{i}\left(t\right)$ must connect to at least one link with other $MCU_{i}\left(t\right)$. All LRN’s are shared and $\sum_{i=1}^{4^{n}}MCU_{i}\left(t\right)$ knows the way to any node in the network, namely everyone knows the topology of the network. This second step has a paradigm neither optimized nor intelligent. So, it is important to employ some tools from swarm intelligence to improve these initial results.

In this way is important to estimate the $LRN_{i}\left(t\right)$ but not all of these nodes are significant to be considered as the best option to establish a link, in which manner we define $LSN_{i}\left(t\right)$ as the list of significant nodes which is a vector that contains the best bandwidths of a certain $MCU_{i}\left(t\right)$. Algorithm 1 shows the methodology to estimate the $LSN_{i}\left(t\right)$ that needs the $i^{th}$MCU and its $LRN_{i}\left(t\right)$. $B_{thr}$ is the threshold bandwidth needed to be considered as significant node, where $\mu_{i}\left(t\right)$ is the statistical mean of bandwidth of the $LRN_{i}\left(t\right)$ while $\sigma_{i}\left(t\right)$ is the population standard deviation of the same list.

Ones $\sum_{i=1}^{\beta }LRN_{i}\left(t_{0}\right)$ is estimated, we take $MCU_{i}\left(t\right)$ as the seed node in order to grow the WSN from this MCU at the instant *t*. Thus, this first approximation takes into account only $\rho_{i}\left(t\right)$ neighbors, where $1\geq \rho_{i}\left(t\right)\geq 3$ and it is calculated by the Equation (Equation 4) as follows:(4)$$B_{thr}=\mu_{i}\left(t\right)-\sigma_{i}\left(t\right).$$
$B_{thr}$ is the threshold of the bandwidths by considering as a significant node the number of the best bandwidths.

**Algorithm 1:** Generator function to estimate the list of significant nodes (LSN) and the number of significant nodes ($\rho_{i}\left(t\right)$).

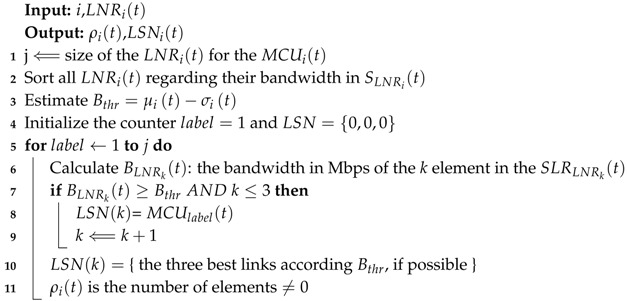



At this moment the WSN can be considered as a scarce network, since only the $\frac{\rho_{i}\left(t\right)}{\beta }$ percent of the reliable links are connected.

### 5.3. Third Stage: Sorting by Means of the Hilbert Fractal Production Rules

The construction of the Hilbert fractal (as shown in Section Hilbert curve L-System) is different by means of the L-system paradigm, based on the paper’s graphic interpretation, being reduced to the replacements of the smallest elements (symbols) or interior replacements. As already explained, in the Hilbert fractal there are copies of $H_{0}$ in $H_{1}$. Each of these copies has to be replaced by a copy of $H_{1}$ to obtain $H_{2}$. However, this is not the only thing that is wanted because the line segments or MCUs that connect must also be modified. With L-systems only formal substitutions are used, so the appropriate size of the line segments is automatically adjusted.

First, the angle π2 radians is chosen. Then $H_{0}$ can be encoded by the symbol *R* (which is interpreted by the parameters to be shared among sensors as +F−F−F+) and $H_{1}$ can be encoded by +RF−LFL−FR+ (see Figure 7). In both cases the set of parameters to be shared leads to the right in the start and end positions. Because $H_{1}$ is obtained from $H_{0}$, the latter determines that the first production rule of the L-system. For the next step, the generation of $H_{2}$, a rewriting rule is needed for *R*. But the only difference between the symbols reflected from *L* and *R* is their orientation. The parameter to be shared traces the *L* curve clockwise—starting with a left turn—whereas the *R* curve counterclockwise, starting with a right turn. Therefore, the rewriting rule of *R* is a mirror of the rule for *L*. Instead of right turns, it makes turns to the left, and vice versa. In addition, the symbols *R* and *L* must be exchanged.

To build the topology of the MCU network, whose goal is to share the parameters among sensors and based on the Hilbert fractal, it was established the following L-system:

L-system:Hilbert curveAxiom:
*R*
Production rules:
R⟶+RF−LFL−FR+


L⟶−LF+RFR+LF−


F⟶F


+⟶−


−⟶+
Parameter:δ=π2 radians

For the first three stages of this L-System n=3, so our adaptation is able to scan $4^{3}=64$ MCUs or $\sum_{i=1}^{4^{3}}MCU_{i}$ gives as a result:

Axiom:
*R*
Stage 1:
−LF+RFR+FL−
Stage 2:
−+RF−LFL−FR+F+−LF+


RFR+FL−F−LF+RFR+FL


−+F+RF−LFL−FR+−
Stage 3:
−+−LF+RFR+FL−F−+RF−LFL−FR+F+RF−LFL−

FR+−F−LF+RFR+FL−+ F + − + RF − LFL − FR + F + −

LF+RFR+FL−F−LF+RFR+FL−+F+RF−LFL−FR


+−F−+RF−LFL−FR+F+−LF+RFR+FL−F−LF+


RFR+FL−+F+RF−LFL−FR+−F+−RL+RFR+FL−


F−+FR−LFL−FR+F+RF−LFL−FR+−F−LF+RFR+FL−+−


When these *n* stages are estimated, the path can be optimized by means of the correlation of the strength of the signal or the bandwidth of the link by the following Equation:(5)$$\phi_{xy}=\frac{\sum_{i=1}^{n}\left(x_{i}-\bar{x}\right)\left(y_{i}-\bar{y}\right)}{\sqrt{\sum_{i=1}^{n}{\left(x_{i}-\bar{x}\right)}^{2}}\sqrt{\sum_{i=1}^{n}{\left(y_{i}-\bar{y}\right)}^{2}}}$$
where $\phi_{xy}$ is the sample correlation coefficient, *n* the stage 0 level of Hilbert curve (Figure 8a), $x_{i}$ and $y_{i}$ are, respectively, the best and the second best individual bandwidths in a certain $MCU_{i}$ indexed inside the WSN as the *i*-th element, while $\bar{x}$ and $\bar{y}$ are, respectively, the sample mean of the best and the second best individual bandwidths in a certain $MCU_{i}$. Thus, we can estimate the best link to a particular $MCU_{i}$.

All $\sum_{i=1}^{4^{n}}MCU_{i}\left(t\right)$ are distributed randomly giving as a result a shape as show in Figure 8b. When γ is reordered as vector, it gives as result the Hilbert Curve Indexing $\vec{\gamma }$ which contains the order of the $MCU_{i}\left(t\right)$, see Figure 8c.

### 5.4. Fourth Stage: Seed Topology of the Growing Wireless Sensor Network

Ones the Hilbert curve is defined as L-system, we adapt the production rules of the original work by David Hilbert [21], who proposed an axiom with a D trajectory, while we propose to start with an U trajectory. Our proposal is based on the fact that the most of the energy is concentrated in the nearest MCU, namely at the right or left. In this way the production rules of the Hilbert curve are defined by:U is changed by LUURL by ULLDR by DRRUD by RDDL.

In this way higher order curves are recursively generated replacing each former level curve with the four later level curves. The Hilbert curve has the property of remaining in an area as long as possible before moving to a neighboring spatial region. Hence, the correlation between neighbor $MCU_{i}\left(t\right)$ is maximized, which is an important property in the optimization of any system. The higher the correlation for estimating the final topology, the more efficient the data parameter sharing.

In this way, using the three first stages, $LRN_{i}\left(t\right)$ is generated, $MCU_{1}\left(t\right)$ is identified, and the production rules are established. According the production rules of the Hilbert curve we use the axiom U.

The use the $LRN_{1}\left(t_{0}\right)$ is intended to choose, if possible, the three best links for shaping the four nodes which are needed to draw a trajectory U, but for the rest of the nodes can be possible to use either U, L, R or D trajectories.

Thus, each node has an initial index in the network which, at the beginning, is not optimized, since it uses a U trajectory. To optimize it, all the elements must be numbered according to the Hilbert fractal L-System and the position where each node passes, replacing and using the Hilbert fractal production rules.

Furthermore, for increasing the density of the WSN, every significant node is considered, at the same time, as a seed of *n* order node and Algorithm 1 is used by the β nodes presented at the instant *t*. So, replacing n−1 orders of the production rules of the Hilbert fractal *n* times, a *n*-th level fractal is estimated if necessary.

Table 1 shows some of the possible parameters that each $MCU_{i}\left(t\right)$ could sense. In this proposal we employed 16 different sensors, but we figured out that the amount of these sensors can increase, so an 8-bit marker is proposed giving as a result that inside the proposed topology we can differentiate 256 types of sensors or WSN devices.

## 6. Results

We have divided the experimental results of this section into five parts:Mainstream topology,Selection of the hub $MCU_{1}\left(t_{0}\right)$,First approximation of the WSN topology based on the Hilbert curve,Distributed learning fractal approximation of the WSN topology based on the Hilbert L-system, andDiscussion of results.

In the first stage the initial considerations are established, whereas in the second one the estimation of the WSN topology is defined. Meanwhile, in the last stage a discussion is carried out in order to compare our results.

### 6.1. Mainstream Topology

For generating a dynamic topology of a certain room, we proposed an evaluation methodology that is subdivided as following:Wireless access point (WAP),WSN $MCU_{i}$,Physical location of $MCU_{i}$.

In this way, we used a Tenda AC6 Wireless WiFi smart router or Tenda-AC1200 (Figure 9) as the hub node of the WSN, which was placed in the center of the floor plan. This WAP had the following features:Supports 802.11ac standard,Guest network access provides secure WiFi access for guests sharing any WSN,Broadcom 900 MHz processor,Generation of WiFi simultaneous 2.4 GHz 300 Mbps and 5 GHz 867 Mbps connections for 1.2 Gbps of total available bandwidth,Network management with Tenda mobile application, andHigh powered amplifiers and four 5 dBi external antennas for whole-room coverage.

Moreover, we used gradually in this example 64 WSN Development Boards divided as follows: 46 boars ModeMCU-ESP8266 and 18 boards Raspberry Pi 3 B+.

Tenda AC1200 router achieved links of 1.2 Gbps using the 5 GHz band, $MCU-RPi3_{i}$ were able to support this standard but $MCU-ESP8266_{i}$ does not support the 802.11ac standard, so we developed a WSN with up to 300 Mbps at 2.4 GHz. It is important to mention we can extend the coverage range if we use a $MCU_{i}$ able to transmit at 5 GHz.

Another statement to point out is we randomly distributed the WSN development boards in two stages. First we located 16 MCUs along the room as we showed it in the Figure 6. Then we added the rest 48 MCUs one by one, in order to optimize the growing of the WSN. In this way, at the instant $t_{0}$, β=16, $WSN_{N}\left(t_{0}\right)$ is defined by $\sum_{i=1}^{\beta =16}MCU_{i}\left(t_{0}\right)$ and *n* = 2.

The Figure 10 gives us an idea about how the devices are placed before calculating the $WSN_{N}\left(t_{0}\right)$ topology and the means can be shared in the sensors connected to each MCU. We define this initial configuration as follows:The WAP was placed in the center of the room, although it can be placed anywhere within the room.Only 16 devices try to connect by the IEEE 802.11 b/g/n protocol to Tenda-AC1200.Those devices or nodes who find Tenda-AC1200 (the hub) were connected to it, whereas those who do not got a link to a central node or router to enable the P2P WiFi mode in listening or waiting mode, see Figure 10.All the nodes connected to the Tenda-AC1200 were completely disabled from the WiFi Direct mode.The WAP sent a broadcasting to the whole network to measure what was the link speed of each node connected to it.There were 16 MNI labels identified. In this case the Hilbert fractal grew up to a third level. In this first generation of MNI’s, all consecutive numbers are assigned according to the speed of its link to Tenda-AC1200.

### 6.2. Selection of the Hub $MCU_{1}\left(t_{0}\right)$

From Figure 10 and Table 2, a NodeMCU-ESP8266 had the best link which is the way this node is labeled as 37. For visual purposes, in this work we refer to this device as MCU-ESP8266-37, although within the $WSN_{N}\left(t_{0}\right)$ network it does not matter what type of device is.

As shown in the Figure 10, our floor plan sample had some walls, which attenuated the signal. That is why certain devices with shorter linear distance have a slower maximum speed than others that are further away from the WPA. Perhaps, MCU-ESP8266-26 was 3.50 m from the router and it can transmit packets at a speed of 195.09 Mbps, while the MCU-ESP8266-43 was 3.75 m but it was linked to 199.39 Mbps, this is due to the vertical wall that divides this room.

The maximum bandwidth of each node or device of the WSN is also shown in Table 2. It can be highlighted that the WAP has linked with sixteen devices and that the highest linked device is the MCU-ESP8266-37 with 225.16 Mbps, and the second highest linked device is the MCU-RPi3-40 with 223.01 Mbps. This process was repeated until completing the sixteen MCUs to be the sufficient linking requirement to transmit.

From Table 2, we estimated a preliminary network topology and thus determined the spatial position of each node. Firstly, it is numbered as node 1 or seed node with label 37, for this example, the MCU-ESP8266-37-1, which is the only one that maintains connection to the WAP, so the $MNI_{i\geq 2}\left(t_{0}\right)$ disable the IEEE 802.11 b/g/n protocol and all the $\sum_{i}^{\beta }MCU_{i}\left(t_{0}\right)$, i.e., from $i=1,2,3\ldots 2^{n}$ enable WiFi Direct mode.

### 6.3. Example of the First Approximation of the Control of the Wireless Sensor Network Based on the Hilbert Curve

Once this was done, each $MCU_{i}\left(t_{0}\right)$ connected to as many $\sum_{i}^{\beta }MCU_{i}\left(t_{0}\right)$ as possible and fills in a list of reliable nodes or $\sum_{i}^{\beta }LRN_{i}\left(t_{0}\right)$. With this, each $MCU_{i}\left(t_{0}\right)$ fills in a list of all the $\sum_{i}^{\beta }MCU_{i}\left(t_{0}\right)$ with which it is linked. In this way we have up to β$LNR_{i}\left(t_{0}\right)$.

In order to show the two possible scenarios:when $\rho_{i}\left(t\right)=3$, andwhen $0>\rho_{i}\left(t\right)>3$

For the first case, we use the $MCU_{49}\left(t_{0}\right)$ as the seed node. Thus for the first example, we can notice in Figure 11 that only five $MCU_{i}\left(t_{0}\right)$ are connected to $MCU_{49}\left(t_{0}\right)$, so its maximum enclosure speed was measured and encapsulated as the $LNR_{49}\left(t_{0}\right)$. Table 3 shows maximum bandwidth of each $MCU_{i}\left(t_{0}\right)$, bold results indicate the two maximum bandwidth for $MCU_{49}\left(t_{0}\right)$, while italics highlight the lowest bandwidth for the same Node.

From this example it can be noted that the maximum link speed was 226.88 Mbps which was reached by the $MCU_{52}\left(t_{0}\right)$ and the slowest connection was 190.79 Mbps obtained by the $MCU_{49}\left(t_{0}\right)$.

In this way and using the Algorithm 1, i=49 so $LRN_{49}\left(t_{0}\right)$ or the 49*^th^* routing table (RT) is estimated, but only some nodes can provide reliability for communication link with the $MCU_{49}\left(t_{0}\right)$. Thus, $LSN_{49}\left(t_{0}\right)$ is the list of significant nodes and a vector that contains the best bandwidths of $MCU_{49}\left(t_{0}\right)$. Also, for estimating $LSN_{49}\left(t_{0}\right)$, it is necessary the $LNR_{49}\left(t_{0}\right)$ of Table 3, in addition according to Equation (Equation 4) in the particular case of $LRN_{49}\left(t_{0}\right)$ the mean is $\mu_{49}\left(t\right)=204.20$ and the standard deviation $\sigma_{49}\left(t\right)=12.06$ giving as a result that $B_{thr}=192.13$. The three best links, if possible, greater or equal to 192.13 were the reliable links: MCU-RPi3-52,MCU-ESP8266-51,MCU-ESP8266-50, namely the number of elements ≠0 in $LSN_{49}\left(t_{0}\right)$ is $\rho_{49}\left(t_{0}\right)=3$.

For the second case, if any label in the $LSN_{i}\left(t\right)$ was zero or the reliable links are $\rho_{i}\left(t\right)<3$ we proposed an alternative solution. That is why we chose $MCU_{37}\left(t_{0}\right)$ as the seed node. In this case, when the Algorithm 1 was employed i=37, $LSN_{37}\left(t_{0}\right)=\;${ 0, *MCU-ESP8266-39*, *MCU-RPi3-40*}, see Figure 12. Whereas $MCU_{37}\left(t_{0}\right)$ was the seed node at the $t_{0}$ instant, the axiom U was employed and it was changed by LUUR to reorder these nodes in a Hilbert fractal shape. At this moment, the WSN can be considered as a scarce network, since only the 4.69% of the reliable links were connected.

### 6.4. Fractal Growing Approximation of Control of the Wireless Sensor Network Based on the Hilbert L-System

For increasing the density of the WSN, every significant node is recursively considered at the same time as a seed node, so Algorithm 1 is used by the 4 nodes presented at the instant $t_{0}$. The axiom U is replaced by (⟶) LUUR using the production rules (Section 5.4) of the Hilbert fractal. In the particular case where it would be need to obtain the third level of the fractal, the alphabet would be recursively replaced up to obtain the L-system as follows:U⟶LUUR-L⟶ULLD*U⟶LUURL⟶ULLDL⟶ULLDD⟶RDDLU⟶LUUR*L⟶ULLDU⟶LUURU⟶LUURR⟶DRRUU⟶LUUR*L⟶ULLDU⟶LUURU⟶LUURR⟶DRRUR⟶DRRU*D⟶RDDLR⟶DRRUR⟶DRRUU⟶LUUR

Figure 13 shows the $MCU_{40}\left(t_{1}\right)$ as the seed at the instant $t_{1}$ MCU by using the axiom R⟶DRRU.

From Figure 14, it is observed, at the instant $t_{2}$, that the $MCU_{36}\left(t_{2}\right)$ is used as the seed MCU by using the axiom R⟶DRRU.

While Figure 15 depicts the $MCU_{35}\left(t_{3}\right)$ as the seed, at the instant $t_{3}$, by using the axiom D⟶RDDL.

In this way the Hilbert fractal rules are iteratively employed up to the instant $t_{j}$.

### 6.5. Discussion of Results

The Hilbert fractal has a closed dimension of two, which is why it is an ordered way to pass any matrix to an ordered vector according to its nearest neighbor and to its speed. This additional parameter gives the sensor network a primary artificial intelligence that optimizes its connection and reconfigures according to the members that are included or disappear from the network. The position inside the WSN of each $MCU_{i}\left(t_{j}\right)$ is showed by Figure 8b, where $t_{j}$ is the last reconfiguration or stabilization of the WSN. For example, we can highlight that the most efficient way to communicate $MNI_{64}\left(t_{j}\right)$ with $MNI_{5}3\left(t_{j}\right)$ is forwarding all the packages to the $MNI_{5}5\left(t_{j}\right)$, then to $MNI_{54}$ to finally reach $MNI_{5}5\left(t_{j}\right)$, instead of just sending the information directly to $MNI_{5}5\left(t_{j}\right)$, see Figure 16. Also in this figure we can highlight the position of each $MCU_{i}\left(t_{j}\right)$ inside this particular floor plan is optimized by the reliable bandwidth. Namely, the best way to communicate any $MCU_{i}\left(t_{j}\right)$ is sending the information to the $MCU_{i+1}\left(t_{j}\right)$ or $MCU_{i-1}\left(t_{j}\right)$. This increases the average bandwidth in the WSN because this set of devices, behaving as a single entity, is able to find an intelligent way to share all the parameters that these devices can measure or estimate. In this way, Figure 16 depicts the half of the floor plan for a visual representation of the ring sorting by applying the production rules of the Hilbert fractal using only the alphabet of second order.

Figure 17 shows an visual representation of the interconnection of all $MCU_{i}\left(t_{j}\right)$, which depicts a double ring, where all the $LRN_{i}\left(t_{j}\right)$ are presented for sharing parameters. In addition, with the usage of the sensor markers, see Table 1, the entire WSN is able to identify what kind of parameter emits a certain $MCU_{i}\left(t_{j}\right)$.

In order to estimate the critical cases, we requested information one thousand times from the $MNI_{62}\left(t_{j}\right)$ to $MNI_{1}\left(t_{j}\right)$ and from $MNI_{16}\left(t_{j}\right)$ to $MNI_{49}\left(t_{j}\right)$, i.e., we chose the farthest possible $MNI_{i}\left(t_{j}\right)$ and we found the following. On average, the proposal can send or receive packages $MNI_{62}\left(t_{j}\right)↔MNI_{1}\left(t_{j}\right)$ with a bandwidth of 208.35 Mbps, while the link $MNI_{16}\left(t_{j}\right)↔MNI_{49}\left(t_{j}\right)$ reach a bandwidth of 207.65 Mbps. The other possible links reach an average of 213.16 Mbps, which is very close to the maximum bandwidth of the IEEE 802.11 b/g/n of 300 Mbps. It is difficult to make a comparison between our proposal against the classical central node configuration since this kind of topology can neither communicate $MNI_{62(t_{j})}↔MNI_{1(t_{j})}$ nor $MNI_{16}\left(t_{j}\right)↔MNI_{49}\left(t_{j}\right)$. In addition, the list of reliable nodes would be only the 16 nodes in Table 2 and the rest of 48 nodes are not able to communicate or request parameters of this central WSN with star topology, even some nodes which are not far each other cannot be directly connected since they have no connection to the WAP. So, star topology has an average of 162.27 Mbps on 16 devices, i.e., 50.89 Mbps or ≈24% less that the average speed of this proposal over the entire WSN.

### 6.6. Impact of Node Density and Quality of Service

In this section we estimated the performance of this proposal DLFA through the simulations of the impact of node density and quality of service (QoS). Also, we estimated the performance of DLFA algorithm in terms of end-to-end delay (E2ED) and packet delivery ratio (PDR). Thus, we compared the performance of DLFA with similar state-of-the-art algorithms:Reliable routing with distributed learning automaton (RRDLA) proposed by Mostafaei in [7].Delay-energy tradeoff in wireless sensor networks with reliable communication (DETR) proposed by Liu et al. in [22].Reliable and energy-efficient routing (REER) proposed by Li et al. in [23].

We use the same network parameters proposed by Zorzi and Rao in [24] and by Karp and Kung in [25] in our experiments.

In this way, it is important to define E2ED as the time it takes a packet to reach its destination after it is generated by its source, which is the sum of its queuing delay and delivery delay [26] while PDR is the probability that the destination node can recover the original packets of a group before the lifetime expires [27]. Based on these definitions, in Equation (Equation 6) we define the η as the ratio of the change between the maximum and the minimum end-to-end delay, to the change between maximum and the minimum packet delivery ratio. If the end-to-end delay is constant, η tends to 0, and if packet delivery ratio is constant η tends to *∞*, regardless of end-to-end delay.
(6)$$\eta \left(\varphi \right)=\frac{E2ED{\left(\varphi \right)}_{\max}-E2ED{\left(\varphi \right)}_{\min}}{PDR{\left(\varphi \right)}_{\max}-PDR{\left(\varphi \right)}_{\min}}$$
where φ refers either the impact of node density or the reliability requirements in terms of QoS. The evaluation setup was taken from the experiment of Mostafaei in [7] according to the models proposed by Niu et al. in [28] and Zeng et al. in [29].

Thus, Figure 18a,b show the comparison between PDR vs E2ED in terms of node density and reliability requirements, respectively. We can highlight that the higher E2ED, the better while the lower PDR, the better. So, both algorithms RRDLA and this proposal obtain the best results and they try to minimize the delay to transfer the event information when destination node can recover the original packets. DLFA is slightly better in both cases, but the differences are not higher than the 2%. In addition, we can notice that the delay both in RRDLA and DLFA remains almost fixed by increasing the node density in the network or the QoS. Both DETR and REER obtain results very far from the two best algorithms.

Table 4 shows η(φ) for the impact of node density and quality of service, we can highlight that in both cases DFLA obtains the best results (in bold) since it tends to 0 and RDDLA gets the second best (in italics).

In conclusion, on the average a packet sent by DFLA takes less time to reach its destination after it is generated by its source increasing the probability that the destination node can recover the original packet before the lifetime expires.

## 7. Conclusions

In this article we used the systemic thinking to develop a DLFA for optimizing a central topology of a WSN, in order to extend the range transmission of a given WiFi network in an intelligent, adaptive and dynamic way when sharing of parameters under a swarm intelligence framework.

In the increasingly interrelated world and due to the global effects of climate change, energy saving is a mandatory task, including the design of any communication protocol based on the WSNs. With the topology of the WSN proposed in the present work, based on the Hilbert fractal—to be a filling-space curve and whose production rules were made up under the L-systems paradigm—and focused on the efficient connection of the MCU, the energy consumption of the connected devices is reduced. Most of this energy consumed by MCUs can be saved through inclusion of new nodes or devices connected in the same WSN, since most of the information detected throughout the network is redundant due to geographically placed sensors. Previously, schemes have already been proposed for the efficient broadcasting of the parameters to be shared within the entire WSN.

However, high communication costs still predominate and the problems shared within the WSNs have not yet been fully solved. That is why in this work we proposed a light broadcasting algorithm in WSNs. Based on the performance analysis, it is possible to verify that the reconfiguration approach of the WSN improves the network’s life time and generates more efficient information of the parameters to be shared by the network, with respect to the traditional schemes. In addition, it is possible to reconfigure the entire WSN at 143.54 μs when only one node was added, and 475.89 μs when more than one node was added up to $4^{n}-1$ devices. Also, it is feasible to obtain some parameters connected to a specific $MCU_{i}$ of the entire WSN in microseconds, which complies with the definition in real time. Finally, it is highlighted that this proposal makes use of swarm intelligence, since each $MCU_{i}$ knows the rest of the nodes of the WSN and knows what happens, even if it is not physically connected, and they understand what happens within the same network, in order to be able to completely reconfigure the topology of the WSN.

This proposal optimizes the link since in a dens network we have 2016 possible links but with only 64 links we can share parameters of sensor, namely with only the 3.17% or we are reducing the 96.83% the number of links in the presented WSN. In addition, we can manage scarse WSNs in order to obtain more dense networks, since by means of a backbone many WAP’s can be interconnected in order to form more complex topologies or more dense, which can grow using higher levels of Hilbert fractal.

Finally, we propose a SDN with strong mobility since it can be reconfigured depending on the amount of nodes, also we employ a target coverage because DLFA only consider reliable links among devices. In terms of reliability, our proposal can share parameters such as battery, radio, hardware, or operating systems. By itself, the quality of service is a challenge that guaranteed level of service delivery to a wireless sensor network, DLFA takes less time to reach its destination after it is generated by its source increasing the probability that the destination node can recover the original packet before the lifetime expires.

## Figures and Tables

**Figure 1 sensors-19-01442-f001:**
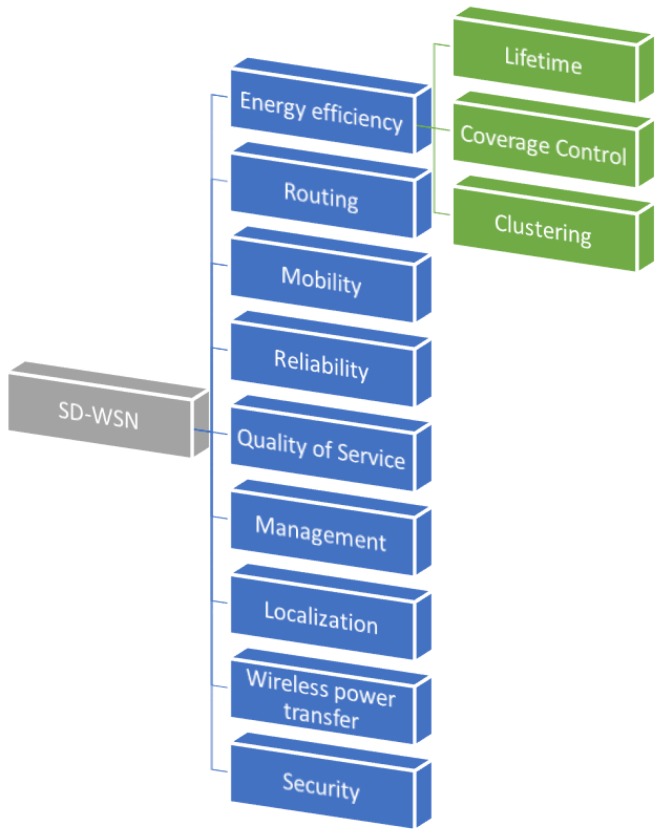
Advances in software-defined wireless sensor networks.

**Figure 2 sensors-19-01442-f002:**
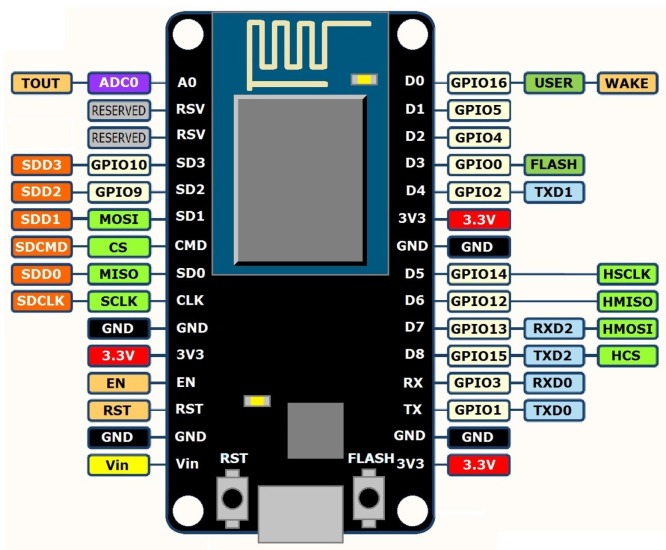
Pin distribution of the WiFi wireless sensor networks (WSNs) development board node micro controller units (MCUs) ESP8266 (MCU-ESP8266) embedded system.

**Figure 3 sensors-19-01442-f003:**
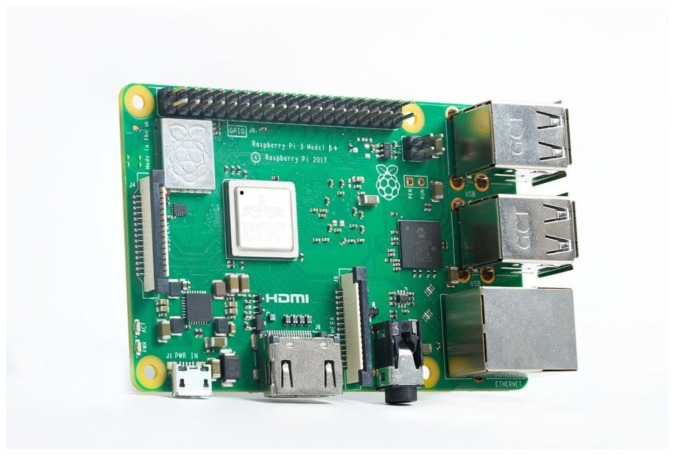
WiFi wireless sensor network (WSN) development board Raspberry Pi 3B+ (MCU-RPi3).

**Figure 4 sensors-19-01442-f004:**
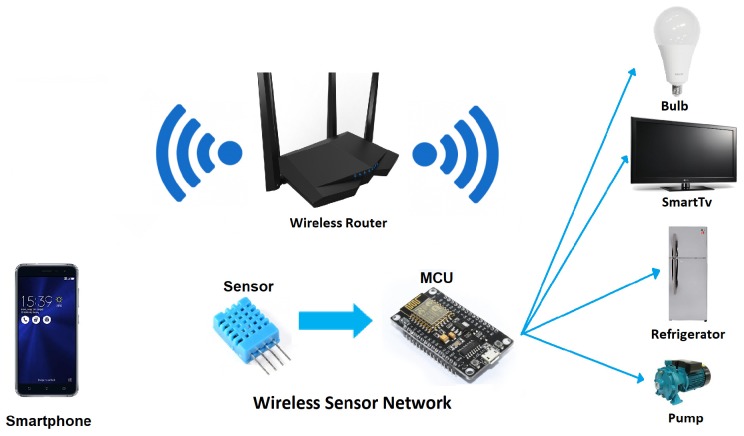
Wi-Fi peer-to-peer and Direct [14].

**Figure 5 sensors-19-01442-f005:**
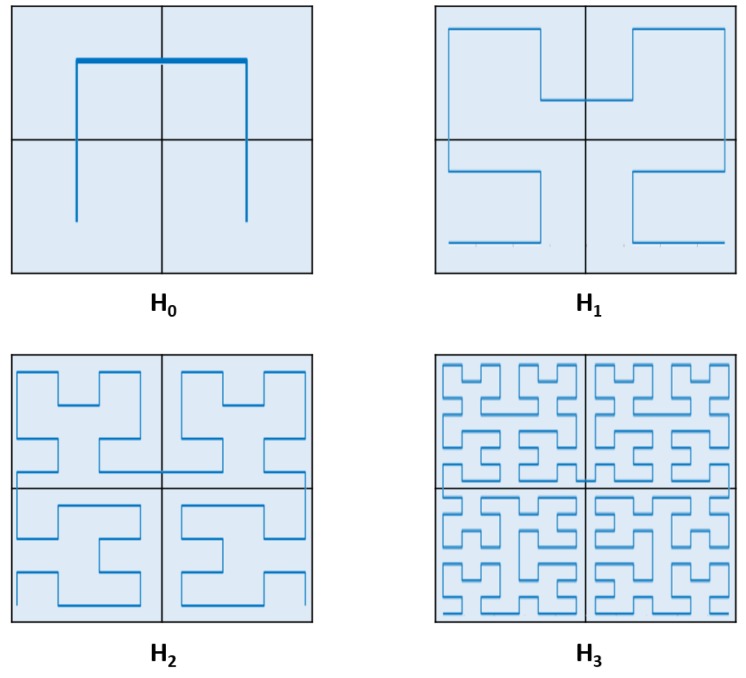
The first four stages of the construction of a Hilbert fractal, axiom = D (see Section 5.3) employed by David Hilbert in [21].

**Figure 6 sensors-19-01442-f006:**
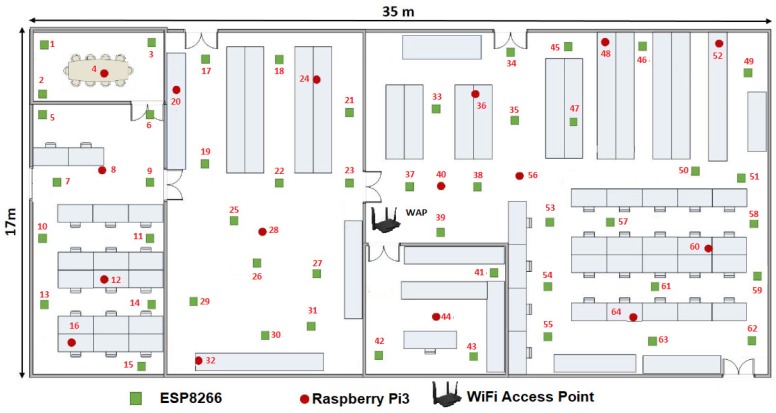
Example of the WSN with swarm intelligence (SI) employed in this work.

**Figure 7 sensors-19-01442-f007:**
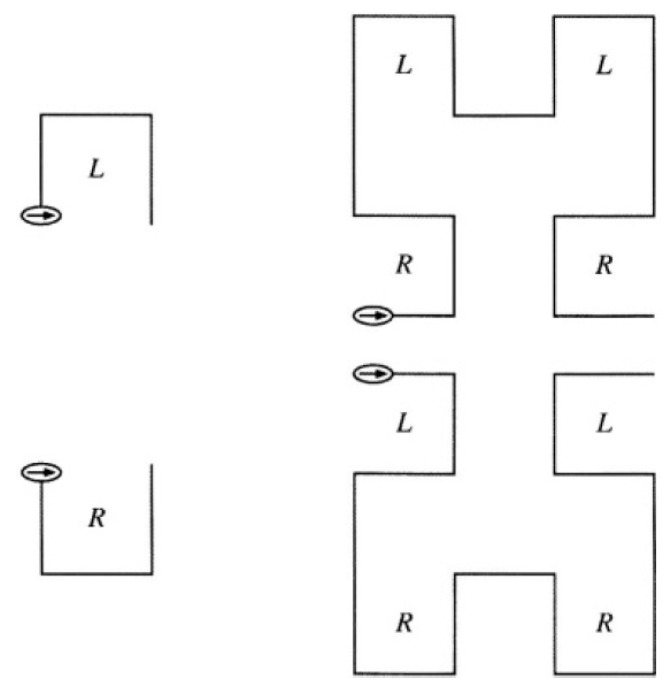
Geometric interpretation of the encoding of the curves and using the symbols L and R (**left**). When starting with axiom R in place of L, we obtain mirror images of the corresponding stages of the Hilbert curve (**right**).

**Figure 8 sensors-19-01442-f008:**
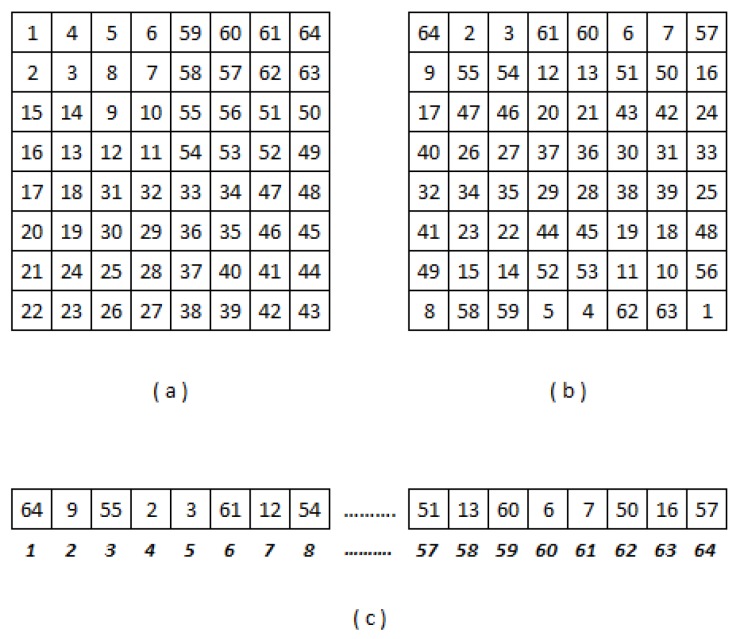
(**a**) First three stages of the Hilbert curve, (**b**) matrix of dispersion or γ of $\sum_{i=1}^{4^{n}}MCU_{i}\left(t\right)$ inside a convectional room, and (**c**) matrix $\vec{\gamma }$ ordered as a vector.

**Figure 9 sensors-19-01442-f009:**
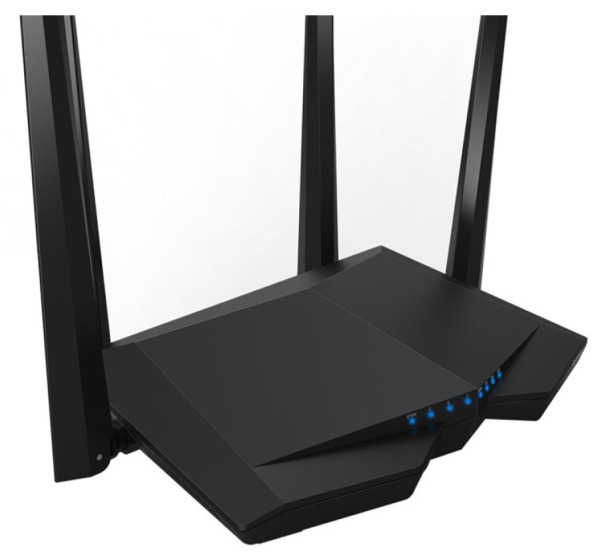
Tenda-AC1200 dual band WiFi smart router.

**Figure 10 sensors-19-01442-f010:**
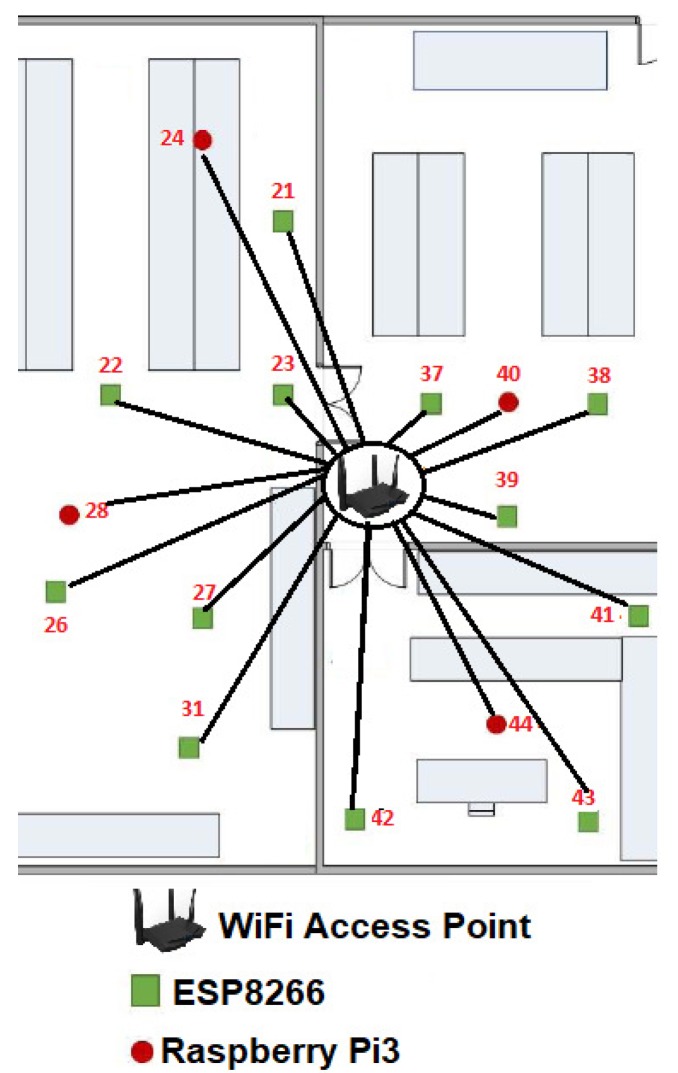
Experiments: reliable $WSN_{N}\left(t_{0}\right)$ development boards by the wireless access point (WAP) Tenda AC1200.

**Figure 11 sensors-19-01442-f011:**
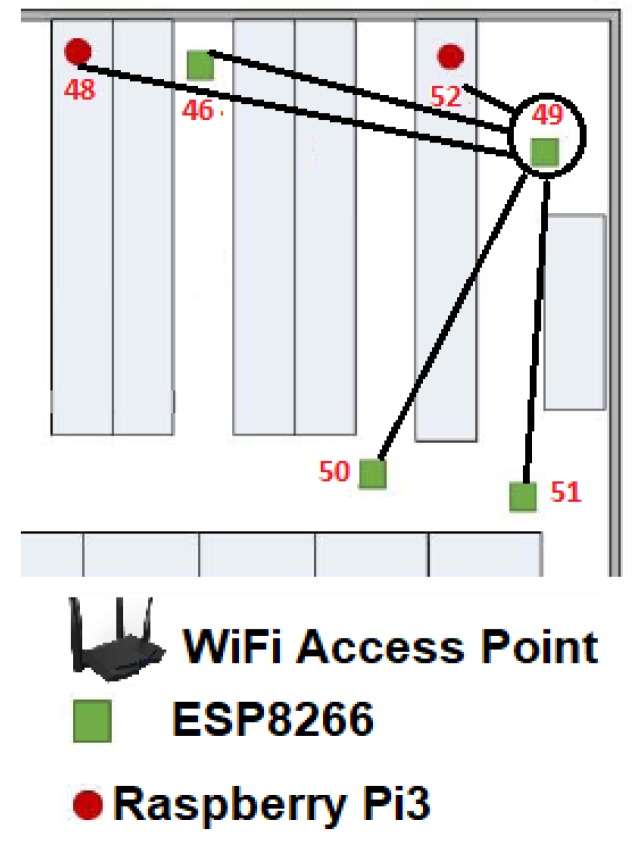
List of reliable nodes of the $MCU_{49}\left(t_{0}\right)$, i.e., $LNR_{49}\left(t_{0}\right)$.

**Figure 12 sensors-19-01442-f012:**
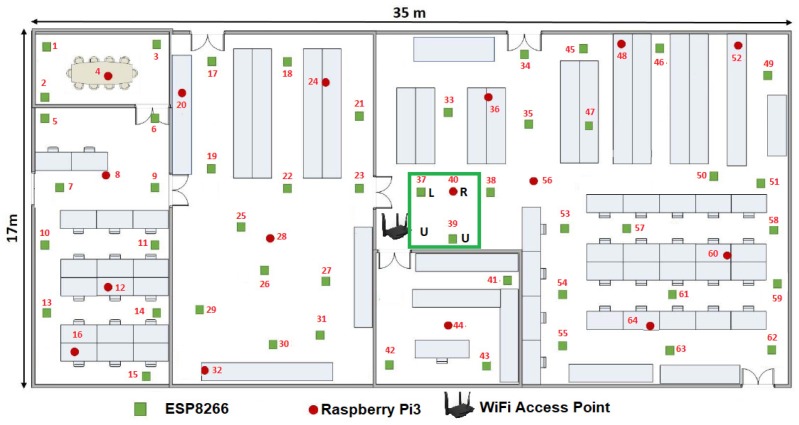
$MCU_{37}\left(t_{0}\right)$ as the seed MCU using the axiom U⟶LUUR.

**Figure 13 sensors-19-01442-f013:**
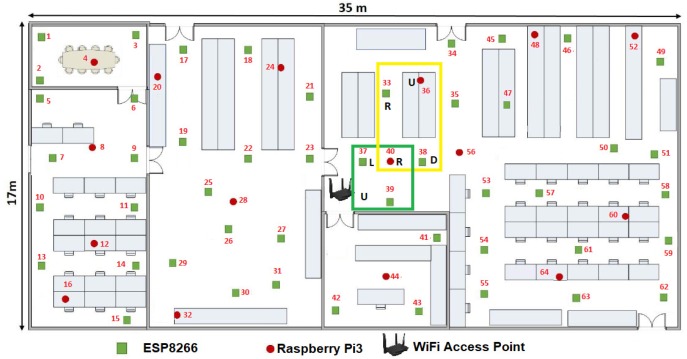
$MCU_{40}\left(t_{1}\right)$ as the MCU seed using the axiom R⟶DRRU.

**Figure 14 sensors-19-01442-f014:**
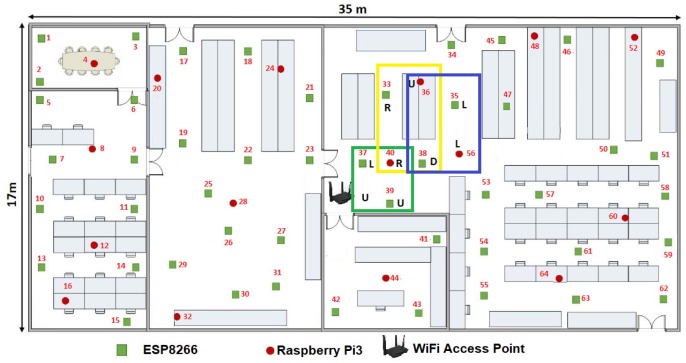
$MCU_{36}\left(t_{2}\right)$ as the MCU seed using the axiom R⟶DRRU.

**Figure 15 sensors-19-01442-f015:**
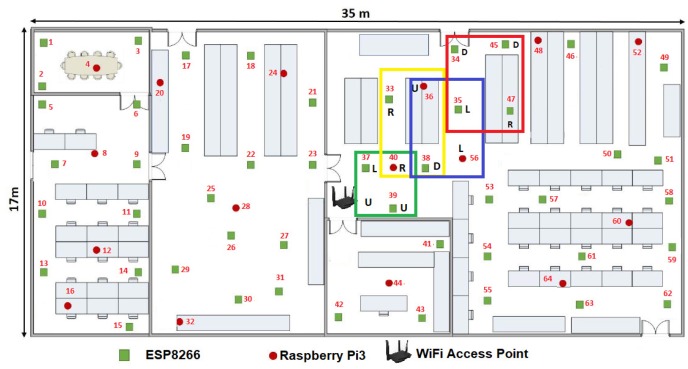
$MCU_{35}\left(t_{3}\right)$ as the MCU using the axiom D⟶RDDL.

**Figure 16 sensors-19-01442-f016:**
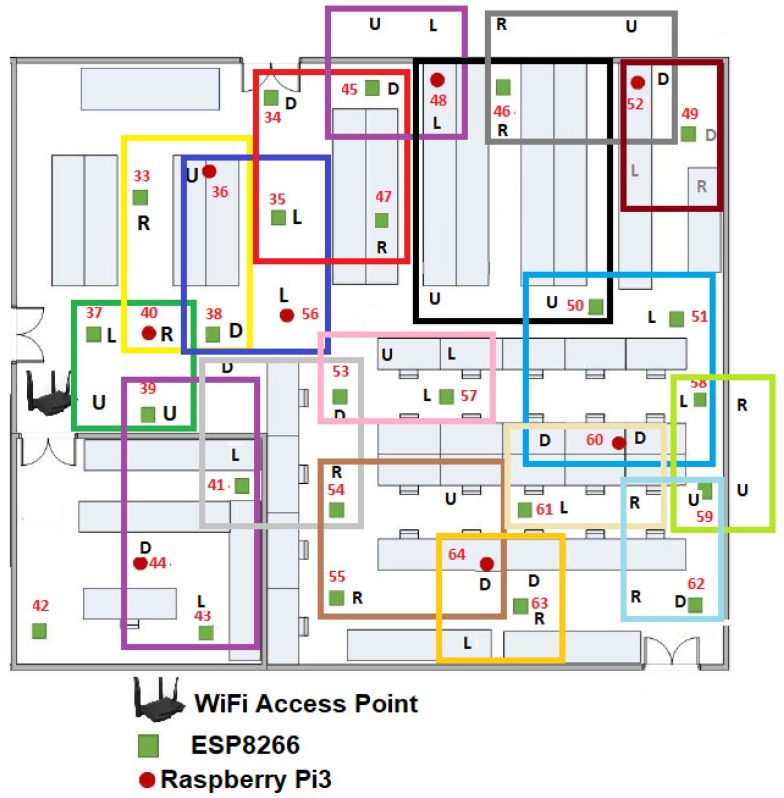
Hilbert fractal used for sorting the half of the reliable members of the proposed WSN applied in a experimental floor plan.

**Figure 17 sensors-19-01442-f017:**
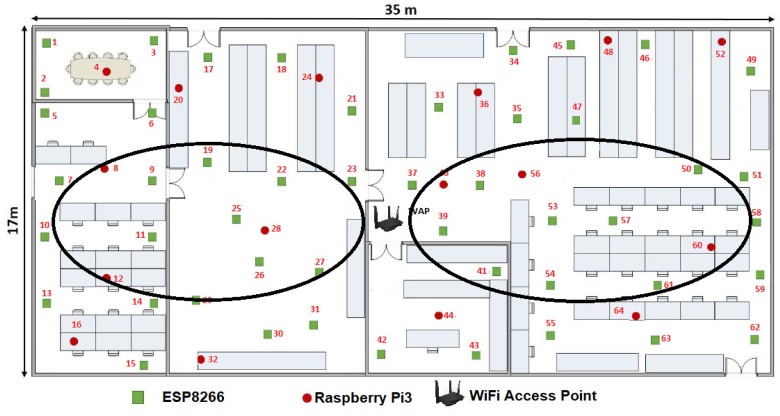
Hilbert fractal used for sorting all the reliable members of the proposed WSN applied in an experimental floor plan.

**Figure 18 sensors-19-01442-f018:**
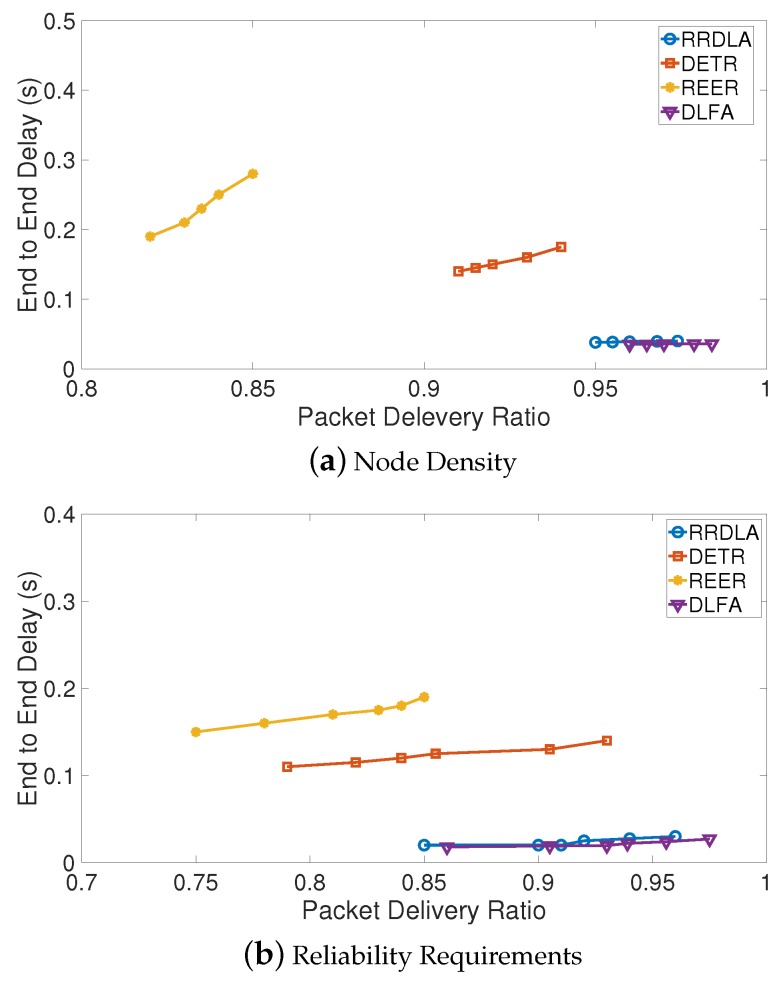
The simulation of the impact of the node density and quality of service, case end-to-end delay vs. packet delivery ratio.

**Table 1 sensors-19-01442-t001:** Some examples of marker of sensor (MS).

00H	Humidity
01H	Temperature
02H	Temperature control
03H	Biosensors
04H	RFID
05H	Biorhythm
06H	Door control
07H	Passive infrared sensor
08H	$CO_{2}$
09H	Web CaM
0AH	Smart Phone
0BH	as sensor
0CH	General purpose MCU
0DH	Pressure sensor
0EH	GPS
0FH	WiFi door bell

**Table 2 sensors-19-01442-t002:** List of the $\sum_{i=1}^{\beta =16}MCU_{i}\left(t_{0}\right)$ devices that have a stable link with the wireless access point (WAP) in Mbps inside the $WSN_{N}\left(t_{0}\right)$ network. The highest linked node is in bold and the second highest linked in italics.

MCU Network Identifier	Linear Distance (m)	Actual Distance (m)	Bandwidth (Mbps)
MCU-ESP8266-21	3.00	3.25	205.83
MCU-ESP8266-22	2.75	2.75	210.12
MCU-ESP8266-23	1.25	1.25	223.01
MCU-RPi3-24	3.50	3.50	203.68
MCU-ESP8266-26	3.50	4.50	195.09
MCU-ESP8266-27	2.50	3.50	203.68
MCU-RPi3-28	3.50	4.00	199.39
MCU-ESP8266-31	4.75	5.75	184.35
**MCU-ESP8266-37**	**1.00**	**1.00**	**225.16**
MCU-ESP8266-38	2.00	2.00	216.57
MCU-ESP8266-39	1.25	1.25	223.01
*MCU-RPi3-40*	*1.25*	*1.25*	*223.01*
MCU-ESP8266-41	3.25	1.25	197.24
MCU-ESP8266-42	3.50	3.50	203.68
MCU-ESP8266-43	3.75	4.00	199.39
MCU-RPi3-44	3.00	3.00	207.98

**Table 3 sensors-19-01442-t003:** $LNR_{49}\left(t_{0}\right)$ or reliable nodes for the MCU-ESP8266-49, **bold** results indicate the two maximum bandwidth for $MCU_{49}\left(t_{0}\right)$, while *italics* highlight the lowest bandwidth for the same node.

MCU Network Identifier	Linear Distance (m)	Actual Distance (m)	Bandwidth (Mbps)
MCU-ESP8266-46	4.00	4.00	199.39
*MCU-RPi3-48*	*5.00*	*5.00*	*190.79*
MCU-ESP8266-50	3.75	3.75	201.53
**MCU-ESP8266-51**	**3.65**	**3.65**	**202.39**
**MCU-RPi3-52**	**0.80**	**0.80**	**226.88**

**Table 4 sensors-19-01442-t004:** η(φ) for the impact of node density and quality of service.

Algorithm	η (Node Density)	η (Quality of Service)
*RDDLA*	*0.0833*	*0.0909*
DETR	1.1667	0.2143
REER	3.000	0.4000
**DFLA**	**0.0417**	**0.0783**

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
