# Peer review of "Distributed Learning Fractal Algorithm for Optimizing a Centralized Control Topology of Wireless Sensor Network Based on the Hilbert Curve L-System"

_sensors, 2019, doi:10.3390/s19061442_

Round 1
Reviewer 1 Report
- The contribution of the paper can be better explained in Introduction section.
- The basic definition for swarm intelligence can be found in several literature. Therefore, it is suitable to assign a full separate section for this purpose. This section provides nothing new to the paper.
- The title for section 3.1 is not clear enough as it starts after section 3 without any explanation.
- There is more need regarding the related work on this area. Some good examples which can be included are as follows; 1) https://doi.org/10.1016/j.jnca.2018.08.012 2)https://doi.org/10.1109/TIE.2018.2869345 3)https://doi.org/10.1016/j.jnca.2018.06.016
- The abstract and conclusion sections can be improved by concentrating on the approach and the results.
- The description of Algorithm 2 is not clear enough. It seems that from lines 1 to 4 performs the initialization of some variables but it is hard to follow from the text.
- Section 6.4 requires more explanation.
Author Response
Thank you for your letter and for the reviewer comments concerning our manuscript entitled Fractal growing model for optimizing a central topology of the IoT sensor network based on the Hilbert Curve L-System with Manuscript ID: sensors-459065 for the publication in Journal Sensors. Those comments are all valuable and very helpful for revising and improving our paper, as well as the important guiding significance to our researches. We have studied comments carefully and have made correction which we hope meet with approval. Revised portion are marked in green for the Reviewer #1 in the paper. The main corrections in the paper and the responds to the reviewer comments and suggestions are in the attached pdf file (Reviewer1.pdf).
We hope that the above answers will satisfy the reviewers and the revised manuscript will be acceptable for publication in Sensors. If anymore explanation or correction is required, we are ready to do that.

Reviewer 2 Report
MicroController Units (MCU’s) should be changed MicroController Units (MCUs). Is there any special reason the symbol is used? If not, the abbreviation should be changed as the described.
One or more experiments are added, then the quality of this paper could be enhanced. Most of all, it can help readers' understanding on the contributions of this research.
Author Response
Thank you for your letter and for the reviewer comments concerning our manuscript entitled Fractal growing model for optimizing a central topology of the IoT sensor network based on the Hilbert Curve L-System with Manuscript ID: sensors-459065 for the publication in Journal Sensors. Those comments are all valuable and very helpful for revising and improving our paper, as well as the important guiding significance to our researches. We have studied comments carefully and have made correction which we hope meet with approval. Revised portion are marked in red for the Reviewer #2 in the paper. The main corrections in the paper and the responds to the reviewer comments and suggestions are in the attached pdf file (Reviewer2.pdf).We hope that the above answers will satisfy the reviewers and the revised manuscript will be acceptable for publication in Sensors. If anymore explanation or correction is required, we are ready to do that.

Round 2
Reviewer 1 Report
The revised version meets professional norms.